# Radiocesium concentrations in mushrooms collected in Kawauchi Village five to eight years after the Fukushima Daiichi Nuclear Power Plant accident

**Limeng Cui[1,2], Makiko Orita[1]\*, Yasuyuki Taira[1], Noboru Takamura[1]**

**1** Department of Global Health, Medicine and Welfare, Atomic Bomb Disease Institute, Nagasaki University Graduate School of Biomedical Sciences, Nagasaki, Japan, **2** Department of Radiation Protection, Beijing Center for Disease Prevention and Control, Beijing Research Center for Preventive Medicine, Beijing, China

\* orita@nagasaki-u.ac.jp

## Abstract

Following the Fukushima Daiichi Nuclear Power Plant accident in March 2011, radionuclides such as iodine-131, cesium-134 and cesium-137 were released into environment. In this study, we collected wild mushrooms from the Kawauchi Village of Fukushima Prefecture, located less than 30 km southwest of the Fukushima nuclear power plant, to evaluate their radiocesium ($^{134}Cs+^{137}Cs$) concentrations and the risk of internal radiation exposure in local residents. 342 mushroom samples were collected from 2016 to 2019. All samples were analysed for radiocesium content by a high–purity germanium detector. Among 342 mushroom samples, 260 mushroom samples (76%) were detected the radiocesium exceeding the regulatory limit of radiocesium (100 Bq/kg for general foods in Japan). The median of committed effective dose from ingestion of wild mushrooms was in the range of 0.015–0.053 mSv in 2016, 0.0025–0.0087 mSv in 2017, 0.029–0.110 mSv in 2018 and 0.011–0.036 mSv in 2019 based on the assumption that Japanese citizens consumed wild mushrooms for 1 year. Thus, our study showed that although radiocesium is still detected in mushrooms collected in Kawauchi village even after 5 to 9 years later, the committed effective dose due to consuming mushrooms was lower than 1 mSv per year. Long-term comprehensive follow-up should monitor radiocesium concentrations in wild mushrooms to support the recovery of the community after the nuclear disaster.

## Introduction

After the Fukushima Daiichi Nuclear Power Plant (FDNPP) accident occurred in March 2011, huge amounts of radionuclides, such as iodine-131 ($^{131}I$), cesium-134 ($^{134}Cs$) and cesium-137 ($^{137}Cs$), were released into the environment. The United Nations Scientific Committee on the Effects of Atomic Radiation (UNSCEAR) estimated that the total releases of $^{131}I$ (8 days of half-life), $^{134}Cs$ (2 years of half-life) and $^{137}Cs$ (30 years of half-life) were 120.0, 9.0 and 8.8 petabecquerels (PBq), respectively [1].

**Funding:** This study was supported by the research projects on health effects of nuclear disasters of the Ministry of the Environment. The funders had no role in study design, data collection and analysis, decision to publish, or preparation of the manuscript.

**Competing interests:** The authors have declared that no competing interests exist.

After the Chernobyl Nuclear Power Plant (CNPP) accident, it has been confirmed that radiocesium tends to concentrate in wild mushrooms [2–9]. Travnikova, et al. found that the average $^{137}$Cs activity in *Boletus luteus*, *B. chanteral*, and *B. russula* wild mushroom samples collected in Veprin village of Russia from 1994 to 1998 were 14,500, 2,550 and 8,980 Bq/kg, respectively [10]. At 30 years after the accident, the median of radiocesium concentrations of *Boletus edulis*, *Leccinum aurantiacum* and *L. scabrum* mushroom samples collected 120 km away from the CNPP were 580, 250 and 290 Bq/kg, respectively [9]. Thus, human consumption of wild mushrooms has contributed greatly to the internal radiocesium body burden around the CNPP [11, 12].

Kawauchi village is located within a 30 km radius from southwest of the FDNPP, and it was the first local authority to have residents return following evacuations after the accident [13]. Currently, almost 80% of residents have returned to the village and restarted their lives. It is customary for residents of some areas of Fukushima Prefecture to eat home-grown plants, such as vegetables, edible wild plants, and mushrooms [14]. Residents living in mountainous areas are more likely to eat home-grown foods than are urban residents. In particular, a common pastime for older adults in the area is to forage for edible wild plants in the mountains or to hunt for wild animals [14]. Before the accident, the village was also famous for collecting wild mushrooms, including *Sarcodon aspratus* and *Tricholoma matsutake*. Because collection and consumption of wild mushrooms is part of the food culture of this village, residents want to know radiocesium levels in the wild mushrooms [15, 16]. National and local governments implemented monitoring of radioactive concentrations over a wide area of Japan in fiscal year (FY) 2016, and 23 out of 2084 wild mushrooms (not mushrooms purchased at the market) tested were found to contain radiocesium that exceeded the standard values [17]. Since 2013, we have evaluated the activity concentration of artificial radiocesium in wild mushrooms collected in Kawauchi village [15, 18]. To support the recovery efforts of the community after nuclear disaster and conduct continuous risk communication with residents, it is important to monitor the radiocesium concentrations in mushrooms not only for regions of Japan, but also in limited areas, such as municipal units, and to minimise internal radiation exposure of residents due to consumption of contaminated foods. The aims of this study were to evaluate the follow-up radiocesium contamination in wild mushrooms collected from 2016 to 2019, which is 5 to 8 years after the accident.

## Materials & methods

### Sampling information

The sampling sites for mushroom collection were in Kawauchi village (the village government office, 37˚ 20′ 15.5″ N, 140˚ 48′ 33.5″ E), 22 km southwest of the FDNPP. After we obtained approval from the Kawauchi Village Office, 342 samples were collected in Autumn from 2016 to 2019. Only approximated locations were recorded because of mushroom points usually kept by villagers as secret for future collection. All samples were classified as saprophytic or symbiotic. Samples of fresh mushroom (14–108 g fresh weight (FW)) were washed and minced and then enclosed in 100 mL plastic containers made of polypropylene for the radionuclide measurements.

### γ−spectrometric measurements

All samples were measured fresh and analyzed for 3,600 s (real time) with a high-purity germanium detector (ORTEC, GMX30–70, ORTEC INTERNATIONAL Inc., Oak Ridge, TN, USA) coupled with a multi-channel analyzer (MCA7600, SEIKO EG&G Co., Ltd., Chiba, Japan). Decay corrections were made based on the sampling date, and detector efficiency calibration

was performed for different measurement geometries using mixed-activity standard volume sources (Japan Radioisotope Association, Tokyo, Japan). The relative efficiency was 31%, and energy resolution of the spectrometer was 1.85 keV for [60]Co. The correction factor of the sum-peak effect of [134]Cs and [137]Cs were almost 1, respectively. Activity concentrations of radiocesium were automatically adjusted based on the date of collection, and the data were defined as the activity concentrations at the collection date. The detection limit of about 11 Bq/kg for [134]Cs and 9 Bq/kg for [137]Cs, the counting errors were ± 6 Bq/kg for [134]Cs (median) and ± 13 Bq/kg for [137]Cs (median), respectively. The radioactivity measurements were performed at Nagasaki University (Nagasaki, Japan). The limit for radiocesium is 100 Bq/kg for general foods. Typically, after the Fukushima Daiichi Nuclear accident, food samples without drying were used for measuring radioactivity.

### Committed effective dose

Internal effective doses from wild mushroom samples were estimated using equation:

$$H_{int} = C \cdot D_{int} \cdot e$$

where $C$ is the activity concentration of detected artificial radiocesium (Bq/kg FW). Here, $D_{int}$ represents the age-dependent dose conversion coefficients presented in International Commission on Radiological Protection (ICRP), Publication 119 [19] for [134]Cs (age 1 years, 1.6E–08 Sv/Bq; age 5 years, 1.3E–08 Sv/Bq; 10 years 1.4E-08 Sv/Bq and age 15–70 years, 1.9E–08 Sv/Bq) and [137]Cs: (age 1 years, 1.2E–08 Sv/Bq; age 5 years, 9.6E-09, age 10 years, 1.0E–08 Sv/Bq; and age 15–70 years, 1.3E–08 Sv/Bq). The value for $e$ was determined from the mean value of daily intake for age and sex which was obtained from the Ministry of Health, Labour, and Welfare, Japan in 2016: males, 2.8–7.3 kg/year; females, 2.5–7.3 kg/year) [20].

### Statistical analysis

Radiocesium concentrations were presented as median, minimum and maximum. Normality was checked by Kolmogorov–Smirnov test. The Jonckheere–Terpstra test was performed to compare radiocesium concentrations in *Sarcodon aspratus* collected during 2016–2019. The Mann–Whitney test was conducted to determine variability between symbiotic and saprophytic species. Data were statistically analyzed using SPSS statistics 25.0 (IBM Corp., Armonk, NY, USA). A p-value less than 0.05 was considered statistically significant. The raw measurement data are provided in S1 File.

### Results

Fig 1 shows the distribution of radiocesium concentrations in wild mushrooms broken down by year and according to concentration distribution. The numbers of wild mushroom samples that exceeded 100 Bq/kg of radiocesium by year were 144 of 192 (75%) in 2016, 10 of 21 (48%) in 2017, 64 of 76 (84%) in 2018 and 42 of 53 (79%) in 2019. Activity concentrations of radiocesium in representative species are shown in Table 1 and activity concentration results for all other species are shown in Supplementary file 2. The median concentrations of [134]Cs were 75 Bq/kg in 2016, n.d. (not detected) in 2017, 92 Bq/kg in 2018 and 25 Bq/kg in 2019, and those of [137]Cs were 454 Bq/kg in 2016, 80 Bq/kg in 2017, 1,006 Bq/kg in 2018 and 339 Bq/kg in 2019, respectively. In addition, the average [134]Cs/[137]Cs ratios were 0.171 in 2016, 0.126 in 2017, 0.091 in 2018 and 0.068 in 2019. Radiocesium concentrations were significantly higher in symbiotic species, such as *Sarcodon aspratus* and *Suillus bovinus*, than saprophytic species, such as *Albatrellus confluens* and *Lyophyllum fumosum* ($p<0.05$).

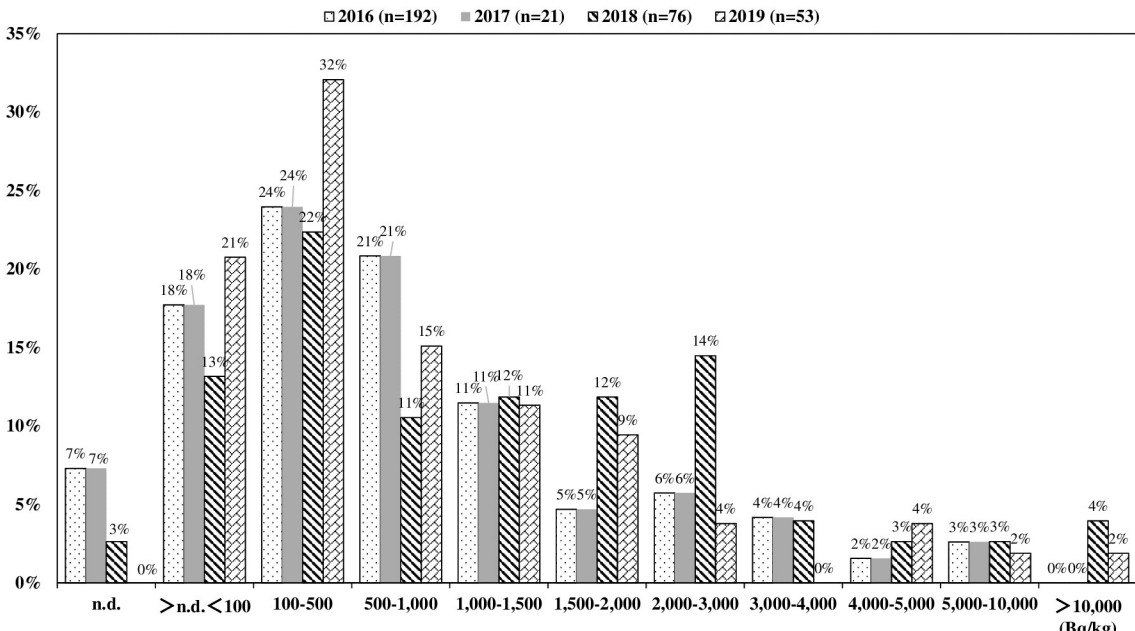

**Fig 1. Distribution of concentrations of radiocesium ($^{134}$Cs+$^{137}$Cs) in wild mushroom samples collected in Kawauchi village (2016–2019).** n.d.: could not be determined.

**Table 1. Activity concentrations of radiocesium (Bq/kg) in mushroom samples (5 species) collected in Kawauchi village from 2016 to 2019.**

| Species | Sample year | n | $^{134}$Cs (Bq/kg) | $^{137}$Cs (Bq/kg) |
|---|---|---|---|---|
| | | | Median (Min-Max) * | Median (Min-Max) |
| *Sarcodon aspratus* | 2016 | 73 | 136 (12–1,079) | 759 (86–6,350) |
| "Kotake" in Japanese | 2017 | 8 | 177 (73–711) | 1,059 (562–1,838) |
| (symbiotic) | 2018 | 40 | 138 (n.d.–1,572) | 1,524 (60–17,155) |
| | 2019 | 21 | 92 (24–1,075) | 1,290 (339–15,867) |
| *Suillus bovinus* | 2016 | 28 | 48 (n.d.–289) | 307 (21–1,774) |
| "Amitake" in Japanese | 2017 | 2 | 15 (n.d.–127) | 74 (21–127) |
| (symbiotic) | 2018 | 3 | 94 (26–219) | 962 (326–2,457) |
| | 2019 | 3 | 27 (n.d.–128) | 336 (53–1,837) |
| *Albatrellus confluens* | 2016 | 20 | n.d. (n.d.–48) | 24 (n.d.–258) |
| "Ningyotake" in Japanese | 2017 | 5 | n.d. | 24 (n.d.–80) |
| (saprophytic) | 2018 | 6 | n.d. | 50 (44–85) |
| | 2019 | 5 | n.d. | 16 (12–55) |
| *Lyophyllum fumosum* | 2016 | 5 | n.d. (n.d.–23) | 39 (27–119) |
| "Shakashimeji" in Japanese | 2017 | 2 | n.d. | 44 (34–53) |
| (saprophytic) | 2018 | 1 | n.d. | 60 |
| | 2019 | 4 | n.d. | 104 (45–158) |
| *Armillaria tabescens* | 2016 | 3 | n.d. | n.d. |
| "Naratake" in Japanese | 2017 | 1 | n.d. | 23 |
| (saprophytic) | 2018 | 1 | n.d. | n.d. |
| | 2019 | 1 | n.d. | 16 |

*Min: minimum; Max: maximum; n.d.: not detected.

A

B

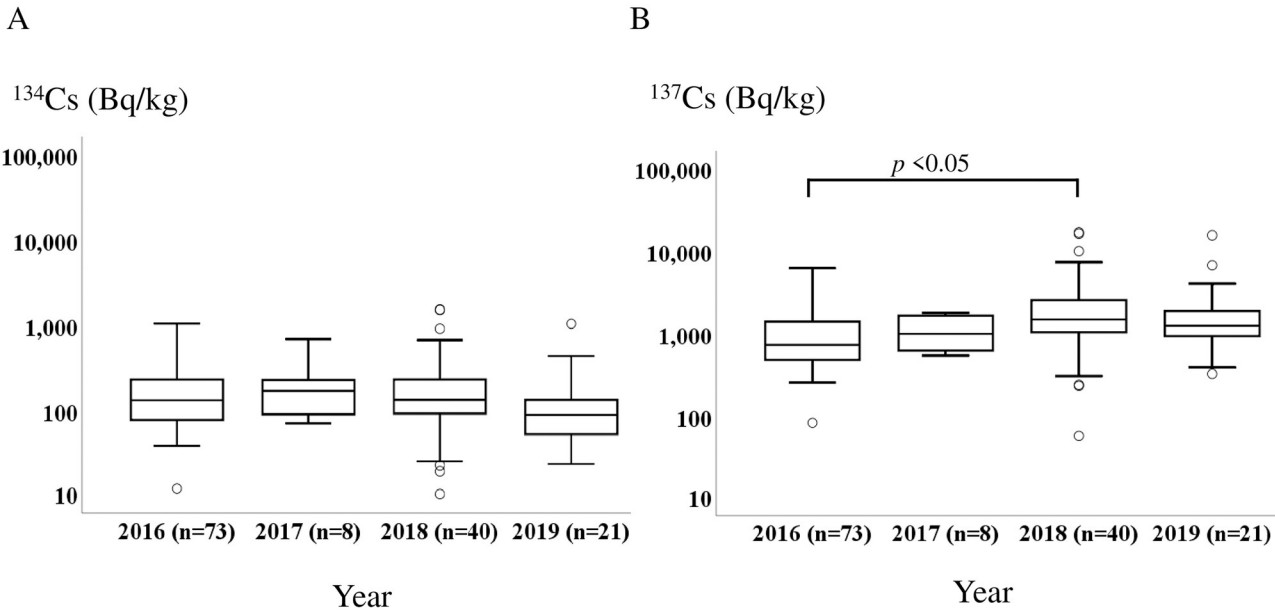

**Fig 2. Boxplots of radiocesium concentrations (a) $^{134}$Cs and (b) $^{137}$Cs in *Sarcodon aspratus* collected from 2016 to 2019.** Significance determined by the Jonckheere–Terpstra test.

We also compared the activity concentrations of radiocesium in *Sarcodon aspratus* samples collected each year (Fig 2). No significant differences were found in $^{134}$Cs activity concentrations ($p > 0.05$). While the activity concentration of $^{137}$Cs in 2018 was significantly higher than that in 2016 (p = 0.004), no significant differences were observed between other years.

Internal committed effective dose for residents were shown in Table 2. The median of committed effective dose from ingestion of wild mushrooms was in the range of 0.015–0.053 mSv in 2016, 0.0025–0.0087 mSv in 2017, 0.029–0.110 mSv in 2018 and 0.011–0.036 mSv in 2019.

## Discussion

After the FDNPP accident, the Fukushima Prefecture governmental office reported monitoring results for radiocesium concentration for agricultural, forestry and fishery products from Fukushima Prefecture. The number of cases exceeding the standard value (100 Bq/kg for general foods) was 419 cases (1.5% of total) in FY2013, 113 cases (0.4% of total) in FY2014, 18 cases (0.08% of total) in FY2015, 6 cases (0.03% of total) in FY2016 and 10 cases (0.05% of total) in FY2017 [21]. We also evaluated radiocesium concentrations in local foods collected in Kawauchi Village. The number of samples exceeding the regulatory radiocesium limit was five out of 4,080 vegetables (0.1%), 652 of 1,986 edible wild plants and fungi (32.8%), and eight of 647 fruits (1.2%). Overall, the concentration of radionuclides in food has decreased with the passage of time [22]. However, radiocesium concentrations in edible wild plants and mushrooms were comparatively higher, while radiocesium was at or near the detection limit for vegetables and fruits.

After the FDNPP accident, several studies have been reported concentrations of radiocesium in wild mushrooms [23–25]. In Namie town, Fukushima Prefecture, the median concentrations of radiocesium in *shiitake* mushroom in each year from fiscal 2012 to fiscal 2015 were 4,070, 973, 2,269 and 742 Bq/kg [25]. Previously, we evaluated radiocesium concentration in wild mushroom samples collected in Kawauchi village and found that radiocesium exceeding

**Table 2. Internal committed effective dose calculated with the assumption that residents consume wild mushrooms collected in Kawauchi village for 1 year.**

| Age (y) | Intake (kg/y)* | Median (minimum—maximum) committed effective dose (mSv/year) | | | | | | | |
|---|---|---|---|---|---|---|---|---|---|
| | | Female | | | | Male | | | |
| | | 2016 | 2017 | 2018 | 2019 | 2016 | 2017 | 2018 | 2019 |
| 1–6 | F: 2.5; M: 2.8 | 0.015 (0.00055–0.31) | 0.0025 (0.00057–0.065) | 0.029 (0.00057–0.58) | 0.011 (0.00061–0.52 | 0.016 (0.00062–0.35) | 0.0027 (0.00063–0.073) | 0.033 (0.00063–0.65) | 0.012 (0.00068–0.58) |
| 7–14 | F: 4.3; M: 4.7 | 0.026 (0.001–0.59) | 0.0039 (0.001–0.12) | 0.052 (0.001–1.1) | 0.019 (0.0011–0.97) | 0.028 (0.0011–0.65) | 0.0043 (0.0011–0.13) | 0.056 (0.001–1.2) | 0.021 (0.0012–1.1) |
| 15–19 | F: 4.2; M: 5.4 | 0.030 (0.0013–0.58) | 0.005 (0.0013–0.12) | 0.063 (0.0013–1.06) | 0.020 (0.0014–0.95) | 0.039 (0.0017–0.75) | 0.0064 (0.0017–0.15) | 0.081 (0.0013–1.3) | 0.026 (0.0018–1.2) |
| 20–29 | F: 5.1; M: 5.1 | 0.037 (0.0016–0.7) | 0.0061(0.0016–0.14) | 0.076 (0.0013–1.3) | 0.025 (0.0018–1.2) | 0.037 (0.0016–0.70) | 0.0061 (0.0016–0.14) | 0.076 (0.0013–1.3) | 0.025 (0.0018–1.2) |
| 30–39 | F: 5.2; M: 4.9 | 0.038 (0.0016–0.72) | 0.0062 (0.0017–0.15) | 0.078 (0.0013–1.3) | 0.025 (0.0018–1.2) | 0.036 (0.0015–0.68) | 0.0058 (0.0016–0.14) | 0.073 (0.0012–1.2) | 0.024 (0.0017–1.1) |
| 40–49 | F: 5.0; M: 5.0 | 0.036 (0.0016–0.69) | 0.006 (0.0016–0.14) | 0.075 (0.0012–1.3) | 0.024 (0.0017–1.1) | 0.036 (0.0016–0.69) | 0.006 (0.0016–0.14) | 0.075 (0.0012–1.3) | 0.024 (0.0017–1.1) |
| 50–59 | F: 6.5; M: 6.7 | 0.047 (0.002–0.9) | 0.0078 (0.0021–0.18) | 0.097 (0.0016–1.6) | 0.032 (0.0022–1.5) | 0.049 (0.0021–0.92) | 0.008 (0.0021–0.19) | 0.1 (0.0017–1.7) | 0.033 (0.0023–1.5) |
| 60–69 | F: 7.3; M: 7.3 | 0.053 (0.0023–1.0) | 0.0087 (0.0023–0.21) | 0.11 (0.0018–1.8) | 0.036 (0.0025–1.7) | 0.053 (0.0023–1.0) | 0.0087 (0.0023–0.21) | 0.11 (0.0023–1.8) | 0.036 (0.0025–1.7) |
| >70 | F: 6.6; M: 6.6 | 0.048 (0.002–0.91) | 0.0079 (0.0021–0.19) | 0.099 (0.0016–1.7) | 0.032 (0.0023–1.5) | 0.048 (0.0021–0.91) | 0.0079 (0.0021–0.19) | 0.099 (0.0021–1.7) | 0.032 (0.0023–1.5) |

* The mean value of daily intake for age and sex from the MHLW. F: Female; M: Male.

100 Bq/kg was detected in 125 of 154 samples (81%) in 2013, and in 123 of 159 samples (77%) in 2015 [15, 18]. In the present study, the percentage of radiocesium in samples exceeding 100 Bq/kg was 75%, 45%, 84% and 79% from 2016 to 2019, respectively. The median concentration of radiocesium in all mushroom species was 526 Bq/kg in 2016, 88 Bq/kg in 2017, 1,106 Bq/kg in 2018 and 362 Bq/kg in 2019. These results suggest that the proportion of mushroom samples with radiocesium concentrations above 100 Bq/kg did not dramatically change in Fukushima over 8 years. Radiocesium concentrations in mushrooms can be affected by many factors of forest, such as species, mycelium habitat and depth, minerals in the soil and contamination level of soil [6, 26–28]. Long-term comprehensive follow-up should be conducted because the existing data is insufficient to characterize risk.

*Sarcodon aspratus* has long been widely collected and consumed in Fukushima. In our previous study, we reported that the median concentration of $^{134}$Cs decreased from 291 Bq/kg in 2013 to 150 Bq/kg in 2015, whereas that of $^{137}$Cs did not decrease (668 Bq/kg in 2013 and 620 Bq/kg in 2015) [15, 18]. In the present study, $^{134}$Cs concentrations of *Sarcodon aspratus* did not significantly decrease from 2016 to 2019. Furthermore, while the activity concentration of $^{137}$Cs in 2018 was significantly higher than that in 2016, no significant differences were observed between other years. The radioactivity of $^{134}$Cs in mushrooms collected in Kawauchi village have reached to relatively low levels and hard to find statistically significant differences from 2016 to 2019, these inconsistencies in changes in $^{137}$Cs concentrations can be considered to be largely due to the relatively long effective half-life of $^{137}$Cs. In the study after the Chernobyl accident, the radiocesium concentrations in mushrooms also showed fluctuation [29, 30]. In Poland, no clear decreasing trend in *C. cibarius* was observed from 1998 to 2013 [29]. In Sweden, the overall concentrations of $^{137}$Cs in mushrooms did not decrease from 1986 to 2007 and increased in *Cantharellus* spp. [30]. In addition, previous studies have made significant contributions to addressing the reasons for the lack of activity reductions over time [28, 31].

Hashimoto S, et al. [31] investigated whether the $^{137}$Cs concentration of wild mushrooms would be spatially biased or influenced by the $^{137}$Cs or exchangeable potassium concentrations in soils because the concentration fluctuates among the same species growing within a given area.

We estimated the committed effective dose based on the average annual intake of mushrooms by Japanese citizens. We showed that the estimated committed effective dose from the only ingestion of wild mushrooms collected in 2016 was 0.015–0.053 mSv, 0.0025–0.0087 mSv in 2017, 0.029–0.110 mSv in 2018 and 0.011–0.036 mSv in 2019 (female and male, median). Previously, we reported that the calculated effective dose ranged from 0.1–1.6 mSv in 2013 [15] and from <0.001 to 0.6 mSv in 2015 [18]. These findings suggest that internal radiation exposure due to the consumption of wild mushrooms remained lower than 1 mSv per year in Kawauchi village. The current reference level (≤1 mSv/year), which was established by the national government, is set at an adequate level such that effects of continued consumption of radioactive material found in agricultural products remain small and safe. The value 1 mSv/year is in accordance with the guidelines set by the Codex Alimentarius Commission, which sets international standards for food [32]. On the basis of food intake according to age group and the effects of radioactive materials on health, the values are set such that the annual additional radiation dose does not exceed approximately 0.9 mSv, even if 50% of foods contain that level of radioactive materials and are ingested continuously. The "limit of 100 Bq kg" for general foods was determined by choosing the most rigorous limits among the calculated values [32]. Our results suggest that the proportion of mushroom samples with radiocesium concentrations above 100 Bq/kg did not dramatically change, but the committed effective dose by consuming mushrooms is estimated to be relatively limited. Experts need to provide careful explanation of the concepts of radiation protection such as the concept of standard values for food.

After the Fukushima accident, we have been cooperating with Kawauchi stakeholders and dialogue with them regarding radiological risk [33]. Our samples were collected in autumn and measured in winter. During regular group meeting in March, we shared results of radioactivity of mushrooms and the committed effective dose by using understandable terms and figures. For the recovery of the community after the nuclear disaster, we suggested that implementing the health promotion of the residents by using the objective data obtained from the local area can contribute to support the decision making of the residents.

Several limitations of this study should be explored. First, sample size was small in 2017. Secondly, although all samples were collected in Kawauchi village, sampling sites were not recorded precisely. Additionally, transfer factors among mushrooms, host plant, soil, organic matter and animals could not be estimated. Uncertainties arise because the committed effective dose from dietary intake of mushrooms does not capture day-to-day variation in individuals. Further, we did not evaluate the potential loss of radiocesium upon cooking in mushrooms. Finally, there is variation in individual consumption of wild mushrooms in Fukushima, and the population has been repeatedly advised not to eat them. It is likely that these are large overestimates for the majority of the population, and underestimates for a small number of individuals who consume large quantities of the most contaminated varieties.

We suggest that providing the basic radioisotope monitoring for wild mushrooms or other foods for human consumption is important for addressing residents' anxiety and for the reduction of future risks of internal exposure.

## Conclusion

We showed that radiocesium in mushrooms collected in Kawauchi village is still detected even 5 to 9 years after the accident and the proportion of mushroom samples with radiocesium

concentrations above 100 Bq/kg did not dramatically change, but the median committed effective dose by consuming mushrooms was found to be lower than 1 mSv per year. We suggest that providing basic radioisotope monitoring inspection for wild mushrooms and other foods for human consumption is important for addressing residents' anxiety and for the reduction of future risks of internal exposure. Long-term comprehensive follow-up should be conducted to clarify the radiocesium concentrations in wild mushrooms in order to support the recovery of the community after the nuclear disaster.

## Supporting information

**S1 File. The original dataset of this study.**
(XLSX)

**S2 File. Radiocesium concentration in mushrooms collected in in Kawauchi village from 2016 to 2019.**
(DOCX)

## Acknowledgments

We would like to thank all the residents and the staff of Kawauchi Village for their cooperation.

## Author Contributions

**Conceptualization:** Limeng Cui, Makiko Orita, Noboru Takamura.

**Data curation:** Limeng Cui.

**Formal analysis:** Limeng Cui, Makiko Orita.

**Funding acquisition:** Noboru Takamura.

**Investigation:** Limeng Cui, Makiko Orita, Noboru Takamura.

**Project administration:** Makiko Orita, Noboru Takamura.

**Supervision:** Yasuyuki Taira.

**Validation:** Noboru Takamura.

**Visualization:** Noboru Takamura.

**Writing – original draft:** Limeng Cui, Makiko Orita.

**Writing – review & editing:** Yasuyuki Taira, Noboru Takamura.

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
