## [Decision Letter · Decision Letter 0]

23 Jul 2020

PONE-D-20-19962

Radiocesium concentrations in mushrooms collected in Kawauchi Village five to eight years after the Fukushima Daiichi nuclear power plant accident

PLOS ONE

Dear Dr. Orita,

Thank you for submitting your manuscript to PLOS ONE. After careful consideration, we feel that it has merit and can be easily edited to meet the journal's needs and  the reviewers suggestions. Therefore, we invite you to submit a revised version of the manuscript that addresses the points raised during the review process. Congratulations!

Both reviewers have generated a small list of helpful suggestions and questions that will help to improve the paper. I encourage you carefully consider these suggestions and incorporate them into the final draft as you see fit. One important suggestion is to include comparisons to measurements in other regions of Japan and other food products (i.e. controls). There is a very large database for food products available for such comparisons. It was also suggested that a little more background nd context be included even though some of this information is available in other published papers. 

We look forward to receiving your revised manuscript.

Kind regards,

Tim A. Mousseau

Academic Editor

PLOS ONE

Journal Requirements:

Reviewers' comments:

Reviewer's Responses to Questions

**Comments to the Author**

1. Is the manuscript technically sound, and do the data support the conclusions?

Reviewer #1: Yes

Reviewer #2: Yes

2. Has the statistical analysis been performed appropriately and rigorously? 

Reviewer #1: Yes

Reviewer #2: Yes

3. Have the authors made all data underlying the findings in their manuscript fully available?

Reviewer #1: Yes

Reviewer #2: Yes

4. Is the manuscript presented in an intelligible fashion and written in standard English?

Reviewer #1: Yes

Reviewer #2: Yes

5. Review Comments to the Author

Reviewer #1: Manuscript Number: PONE-D-20-19962

Manuscript Title: Radiocesium concentrations in mushrooms collected in Kawauchi Village five to eight years after the Fukushima Daiichi nuclear power plant accident

Overall comments:

This manuscript describes interesting new research that is an important part of the continuing understanding of recovery from the Fukushima meltdowns. This reviewer strongly recommends publication but with minor revisions below.

This work would be improved by adding data for pre-meltdown activities of these same species, as well as control samples of mushrooms from areas not affected by this contamination event. The lack of control and background samples is a notable ommission. Likewise, there is no comparison to other locally-grown or regionally-available foods or edible plants, or even staples such as rice and grains. This deficiency applies equally to activity and committed effective dose data.

Specific comments

L21-28: Quantitative data supports that 76% of samples exceed the allowable limit of 100 Bq/kg. By comparison the committed effective dose is described as relatively low but is not compared to a regulatory limit nor is there a comparison to other parts of the diet of returned residents of Kawauchi Village.

Do wild mushrooms have more or less activity compared to other local foods?

If all foods in the villagers’ diet had the same activity as these mushrooms, is that acceptable based on human health considerations?

Which conclusion is dominant – should mushrooms be avoided because they exceed 100 Bq/kg or can they be eaten because effective committed doses are relatively low?

L17 and also L44: Authors give distances from Fukushima, but should give direction as well. Replace “X km from nuclear plant” with “X km south of nuclear plant” as appropriate.

L41: Replace “concentration” with “activity” wherever appropriate in the manuscript. Remove unnecessary “of” from this line as well.

L61: The introduction ends abruptly, without giving much context or discussion of prior work with this study. Has this author or any other researcher also studied edible mushrooms in the region? How common is wild mushroom consumption?

L72: Samples were analyzed by wet weight, without drying. Is the general food standard of 100 Bq/kg also a wet weight standard or a dry weight standard? The authors should clarify this in the text.

L158: What is the purpose of the proposed long-term follow-up? Is it because the existing data is insufficient to characterize risk, or is it because there is not yet a documented reduction in percentage of mushrooms exceeding the general food standard?

L162 to 173: The authors imply that the only removal mechanism for Cs-137 and Cs-134 from village soils is radioactive decay. The authors should explicitly state whether this is the only removal mechanism, and that erosion, remediation, infiltration to lower soil depths and so on do (or do not) take place. The authors correctly note that other researchers have done significant work to address these questions, and readers would be interested if other researchers investigated the reasons for the lack of activity reductions over time.

L176: The committed effective dose from mushrooms is important data, but it needs context. What is the committed effective dose (on average) from the whole diet? What would be the committed effective dose if all foods eaten were at 100 Bq/kg? Are these doses higher or lower than for other foods? How do mushrooms compare to the activity of other local foods?

L180: The overall quality of the written text is excellent, but errors of tense (‘has’ used instead of the correct ‘have’) mar the otherwise logical and readable nature of this manuscript.

L181: Including the authors’ communications with stakeholders is something that is not done nearly enough by other researchers. This is a commendable effort by the authors and I applaud them for it.

L198: Again, the use of nonquantitative terms such as ‘relatively-limited’ should be buttressed with quantitative data. The previous comments for L21-28 should also be reflected in the conclusions.

Reviewer #2: This is a useful paper which updates findings from a long-term study reported in previous papers, such as Nakashima et al, 2015, Orita et al 2017, Tsuchiya et al 2017. The data is clearly presented overall. Some pertinent background information that would give useful context, and was generally included in the earlier papers, is absent. Specifically, it would be helpful to note relevant aspects of food testing policy in Fukushima overall, such as the fact that wild gathered food items such as these mushrooms are rarely included in official testing. The range of Cs activities reported by official testing for agriculturally grown and commercially distributed mushrooms could be given for context. Also, it is important to note that although the Fukushima public has been repeatedly advised since 2011 to avoid eating wild mushrooms because of contamination risk, many people, particularly elderly, do in fact continue to eat them. Helping to clarify the risks to such people is, I believe, the underlying motivation for this ongoing study in Kawauchi.

Comments regarding specific sections are listed below by text line number:

50) “Before the accident, the village was famous for collecting wild mushrooms, including Sarcodon aspratus and Tricholoma matsutake.”

Comment: In Tsuchiya et al, 2017, you provided common Japanese names for the mushrooms in addition to Latin names. This is helpful for non-botanists as well as researchers in the social sciences and others who may find interest in this study.

+++++

93) “The value for e was determined from the mean value of daily intake for age and sex which was obtained from the Ministry of Health, Labour, and Welfare, Japan in 2016: males, 2.8–7.3 kg/year; females, 2.5–7.3 kg/year) [18].”

Comment: There are a few issues that should be clarified regarding these estimates and how they are derived:

— From the citation [18] it appears that the reference document is: The National Health and Nutrition Survey in Japan, 2016 (direct link https://www.mhlw.go.jp/bunya/kenkou/eiyou/dl/h28-houkoku.pdf). If this is the case, then it should be cited directly, and relevant pages should be cited as well.

— It is not clear that all of the mushroom varieties examined in this study are included in this MHLW document. Further, it is not clear that “wild” versus “commercially distributed” mushrooms are categorized separately by MHLW.

— MHLW gives fairly detailed breakdown of consumption quantity estimates by age. It is not clear how this is factored into the value of e obtained, in conjunction with the age-dependent dose conversion coefficients. Is only the mean of all age groups used?

— The larger question is whether or not MHLW’s estimates for typical mushroom consumption are accurate in the case of wild mushrooms gathered and eaten in Fukushima.

++++

119) "Fig 1. Distribution of concentrations of radiocesium(134Cs+137Cs) in wild mushroom samples collected in Kawauchi village (2016–2019)."

Comment: The activities are separated into only four bins: n.d; n.d.-100Bq; 100-1000Bq; >1000Bq. This appears consistent with the earlier papers, although Nakashima et al, 2015, uses five bins. It would be helpful to include more granular histogram for each year, with at least ten bins each, so the actual activity distributions could be better grasped.

++++

141) "Table 2. Internal committed effective dose calculated with the assumption that residents consume wild mushrooms collected in Kawauchi village for 1 year."

Comment: Related to the above comment. It would be helpful to add the estimated quantity consumed for each age group to the table.

++++

152) "The median concentration of radiocesium in all mushroom species was 526 Bq/kg in 2016, 88 Bq/kg in 2017, 1,106 Bq/kg in 2018 and 362 Bq/kg in 2019."

Comment: These are very wide fluctuations. You touch on some of the reasons for this later. Perhaps one finding is that the sampling methodology reported in this study is unable to provide representative measurements of the overall change in radiocesium levels in wild mushrooms in general in Fukushima. However, it can provide a good indication of typical intake of radiocesium from consumption of gathered wild mushrooms.

++++

154) "These results suggest that the proportion of mushroom samples with radiocesium concentrations above 100 Bq/kg did not dramatically change in Fukushima over 8 years."

Comment: This is a key finding, and consistent with the literature regarding Chernobyl.

++++

75) "We showed that the estimated committed effective dose from the ingestion of wild mushrooms collected in 2016 was 0.017 – 0.054 mSv, 0.0028 – 0.0087 mSv in 2017, 0.034 – 0.110 mSv in 2018 and 0.011 – 0.036 mSv in 2019. Previously, we reported that the calculated effective dose ranged from 0.1 – 1.6 mSv in 2013 [14] and from <0.001 to 0.6 mSv in 2015 [16]. These findings suggest that internal radiation exposure due to the consumption of wild mushrooms remained relatively limited in Kawauchi village."

Comment: These points as stated appear to be valid. However, as noted later, there is great variation in individual consumption of wild mushrooms in Fukushima, and the population has been repeatedly advised not to eat them. It is likely that these are large overestimates for the majority of the population, and underestimates for a small number of individuals who consume large quantities of the most contaminated varieties.

++++

190) "Uncertainties arise because the committed effective dose from dietary intake of mushrooms does not capture day-to-day variation in individuals."

Comment: As noted above.

6. PLOS authors have the option to publish the peer review history of their article (what does this mean?). If published, this will include your full peer review and any attached files.

Reviewer #1: **Yes: **Marco Kaltofen, PhD.

Reviewer #2: No

---

## [Author Response · Author response to Decision Letter 0]

31 Aug 2020

Reviewer #1:

Overall comments:

This manuscript describes interesting new research that is an important part of the continuing understanding of recovery from the Fukushima meltdowns. This reviewer strongly recommends publication but with minor revisions below. This work would be improved by adding data for pre-meltdown activities of these same species, as well as control samples of mushrooms from areas not affected by this contamination event. The lack of control and background samples is a notable ommission. Likewise, there is no comparison to other locally-grown or regionally-available foods or edible plants, or even staples such as rice and grains. This deficiency applies equally to activity and committed effective dose data.

We greatly appreciate your comments on our manuscript. We have revised some sections and added more information on comparison values. Although we could not obtain data on the radioactive material concentration in mushrooms in Kawauchi Village before the accident, we added the food monitoring data in other regions and other food products after the Fukushima accident for comparison in Discussion section. (Page 10, lines 159-169)

Specific comments

L21-28: Quantitative data supports that 76% of samples exceed the allowable limit of 100 Bq/kg. By comparison the committed effective dose is described as relatively low but is not compared to a regulatory limit nor is there a comparison to other parts of the diet of returned residents of Kawauchi Village.

According to your suggestion, we revised the Discussion section as follows; “These findings suggest that internal radiation exposure due to the consumption of wild mushrooms remained lower than 1 mSv per year in Kawauchi village. The current reference level (1 mSv/year), which was established by the national government, is set at an adequate level such that effects of continued consumption of radioactive material found in agricultural products remain small and safe. The value 1 mSv/year is in accordance with the guidelines set by the Codex Alimentarius Commission, which sets international standards for food [32]. On the basis of food intake according to age group and the effects of radioactive materials on health, the values are set such that the annual additional radiation dose does not exceed approximately 0.9 mSv, even if 50% of foods contain that level of radioactive materials and are ingested continuously. The “limit of 100 Bq kg” for general foods was determined by choosing the most rigorous limits among the calculated values [32]. Our results suggest that the proportion of mushroom samples with radiocesium concentrations above 100 Bq/kg did not dramatically change, but the committed effective dose by consuming mushrooms is estimated to be relatively limited. Experts need to provide careful explanation of the concepts of radiation protection such as the concept of standard values for food.” (Page 12, Line 208-222.)

Do wild mushrooms have more or less activity compared to other local foods?

If all foods in the villagers’ diet had the same activity as these mushrooms, is that acceptable based on human health considerations?

We added food monitoring data from other regions and for other food products after the Fukushima accident for comparison in the Discussion section. In addition, we considered concepts of radiation protection, especially for consideration of the standard values for food products. (Page 10, Lines 159-169 and Page 12, Lines 208-222)

Which conclusion is dominant – should mushrooms be avoided because they exceed 100 Bq/kg or can they be eaten because effective committed doses are relatively low?

There is variability in individual consumption of wild mushrooms in Fukushima, and the population has been repeatedly advised not to eat them. It is likely that these values are greatly overestimated for the majority of the population and underestimated for a small number of individuals who consume large quantities of the most contaminated varieties. We suggest that providing basic radioisotope monitoring inspection for wild mushrooms and other foods for human consumption is important. We revised the limitation of this study and Conclusion section. (Pages 13, Lines 235-242, and 245-253.)

L17 and also L44: Authors give distances from Fukushima, but should give direction as well. Replace “X km from nuclear plant” with “X km south of nuclear plant” as appropriate.

According to your suggestion, we added the direction from the Fukushima daiichi nuclear power plant in the Abstract and Introduction section. We also describe the location as being within a set radius of the plant (Lines 17 and 49.)

L41: Replace “concentration” with “activity” wherever appropriate in the manuscript. Remove unnecessary “of” from this line as well.

We replace “concentration of” with “activity” on Page 3, Line 42.

L61: The introduction ends abruptly, without giving much context or discussion of prior work with this study. Has this author or any other researcher also studied edible mushrooms in the region? How common is wild mushroom consumption?

We added this information in Introduction session as follows; “It is customary for residents of some areas of Fukushima Prefecture to eat home-grown plants, such as vegetables, edible wild plants, and mushrooms [14]. Residents living in mountainous areas are more likely to eat home-grown foods than are urban residents. In particular, a common pastime for older adults in the area is to forage for edible wild plants in the mountains or to hunt for wild animals [14]. Before the accident, the village was also famous for collecting wild mushrooms, including Sarcodon aspratus and Tricholoma matsutake. Because collection and consumption of wild mushrooms is part of the food culture of this village, residents want to know radiocesium levels in the wild mushrooms [15, 16]. National and local governments implemented monitoring of radioactive concentrations over a wide area of Japan in fiscal year (FY) 2016, and 23 out of 2084 wild mushrooms (not mushrooms purchased at the market) tested were found to contain radiocesium that exceeded the standard values [17]. Since 2013, we have evaluated the activity concentration of artificial radiocesium in wild mushrooms collected in Kawauchi village [15, 18]. To support the recovery efforts of the community after nuclear disaster and conduct continuous risk communication with residents, it is important to monitor the radiocesium concentrations in mushrooms not only for regions of Japan, but also in limited areas, such as municipal units, and to minimise internal radiation exposure of residents due to consumption of contaminated foods. The aims of this study were to evaluate the follow-up radiocesium contamination in wild mushrooms collected from 2016 to 2019, which is 5 to 8 years after the accident.” (Page 2 and 3, Lines 51-69.)

L72: Samples were analyzed by wet weight, without drying. Is the general food standard of 100 Bq/kg also a wet weight standard or a dry weight standard? The authors should clarify this in the text.

According to your suggestion, we added this clarification to the Materials and methods as follows: “The limit for radiocesium is 100 Bq/kg for general foods. Typically, after the Fukushima Daiichi Nuclear accident, food samples without drying were used for measuring radioactivity.” (Page 5, Line 94-95.)

L158: What is the purpose of the proposed long-term follow-up? Is it because the existing data is insufficient to characterize risk, or is it because there is not yet a documented reduction in percentage of mushrooms exceeding the general food standard?

According to your suggestion, we revised the sentence as follows, “Long-term comprehensive follow-up should be conducted because the existing data is insufficient to characterize risk.” (Pages 11, Lines 183-184.)

L162 to 173: The authors imply that the only removal mechanism for Cs-137 and Cs-134 from village soils is radioactive decay. The authors should explicitly state whether this is the only removal mechanism, and that erosion, remediation, infiltration to lower soil depths and so on do (or do not) take place. The authors correctly note that other researchers have done significant work to address these questions, and readers would be interested if other researchers investigated the reasons for the lack of activity reductions over time.

We added the following explanation to the Discussion section: “previous studies have made significant contributions to addressing the reasons for the lack of activity reductions over time [28, 31]. Hashimoto S, et al. [31] investigated whether the 137Cs concentration of wild mushrooms would be spatially biased or influenced by the 137Cs or exchangeable potassium concentrations in soils because the concentration fluctuates among the same species growing within a given area.” (Page 11, Lines 198-202.)

L176: The committed effective dose from mushrooms is important data, but it needs context. What is the committed effective dose (on average) from the whole diet? What would be the committed effective dose if all foods eaten were at 100 Bq/kg? Are these doses higher or lower than for other foods? How do mushrooms compare to the activity of other local foods?

We estimated the committed effective dose from the ingestion of wild mushrooms alone. According to your suggestion, we revised the Discussion section including the concepts of radiation protection, especially, and the standard values for food and comparison with the activity of other local foods.

L180: The overall quality of the written text is excellent, but errors of tense (‘has’ used instead of the correct ‘have’) mar the otherwise logical and readable nature of this manuscript.

We replace “has” with “have” on Page 12, Line 223.

L181: Including the authors’ communications with stakeholders is something that is not done nearly enough by other researchers. This is a commendable effort by the authors and I applaud them for it.

Thank you for your comment.

L198: Again, the use of nonquantitative terms such as ‘relatively-limited’ should be buttressed with quantitative data. The previous comments for L21-28 should also be reflected in the conclusions.

According to your suggestion, we added specific values in the Conclusion as follows: “the median committed effective dose by consuming mushrooms was found to be lower than 1 mSv per year.” (Page 13, Line 248-249)

 

Reviewer #2: 

This is a useful paper which updates findings from a long-term study reported in previous papers, such as Nakashima et al, 2015, Orita et al 2017, Tsuchiya et al 2017. The data is clearly presented overall. Some pertinent background information that would give useful context, and was generally included in the earlier papers, is absent. Specifically, it would be helpful to note relevant aspects of food testing policy in Fukushima overall, such as the fact that wild gathered food items such as these mushrooms are rarely included in official testing. The range of Cs activities reported by official testing for agriculturally grown and commercially distributed mushrooms could be given for context. Also, it is important to note that although the Fukushima public has been repeatedly advised since 2011 to avoid eating wild mushrooms because of contamination risk, many people, particularly elderly, do in fact continue to eat them. Helping to clarify the risks to such people is, I believe, the underlying motivation for this ongoing study in Kawauchi.

We greatly appreciate your comments on our manuscript. We have changed some sections and added more information.

0) “Before the accident, the village was famous for collecting wild mushrooms, including Sarcodon aspratus and Tricholoma matsutake.”

Comment: In Tsuchiya et al, 2017, you provided common Japanese names for the mushrooms in addition to Latin names. This is helpful for non-botanists as well as researchers in the social sciences and others who may find interest in this study.

According to your suggestion, we added Japanese names of mushrooms in Table 1. (Page 7)

93) “The value for e was determined from the mean value of daily intake for age and sex which was obtained from the Ministry of Health, Labour, and Welfare, Japan in 2016: males, 2.8–7.3 kg/year; females, 2.5–7.3 kg/year) [18].”

Comment: There are a few issues that should be clarified regarding these estimates and how they are derived:

— From the citation [18] it appears that the reference document is: The National Health and Nutrition Survey in Japan, 2016 (direct link https://www.mhlw.go.jp/bunya/kenkou/eiyou/dl/h28-houkoku.pdf). If this is the case, then it should be cited directly, and relevant pages should be cited as well.

We revised reference 20 for research the quoted the part directly, “Ministry of Health, Labour and Welfare, Japan. Report on National Health and Nutrition Survey, pages 72-73 and 76-77. (in Japanese) [cited August 31, 2020]. In: https://www.mhlw.go.jp/bunya/kenkou/eiyou/dl/h28-houkoku-04.pdf”. (Page 20)

— It is not clear that all of the mushroom varieties examined in this study are included in this MHLW document. Further, it is not clear that “wild” versus “commercially distributed” mushrooms are categorized separately by MHLW.

The MHLW results show the total mushroom intake in the population. The MHLW data contains all kinds of mushrooms (both wild and commercially distributed). We added this as one of the limitations of this study. (Page 13, Line 235-242).

— MHLW gives fairly detailed breakdown of consumption quantity estimates by age. It is not clear how this is factored into the value of e obtained, in conjunction with the age-dependent dose conversion coefficients. Is only the mean of all age groups used?

We calculated the internal effective doses using the median intake of each age group and the age-dependent dose conversion coefficients. We added the mushroom intake information for each age group in Table 2.

— The larger question is whether or not MHLW’s estimates for typical mushroom consumption are accurate in the case of wild mushrooms gathered and eaten in Fukushima.

As you pointed out, we are not sure whether the typical MHLW estimates of mushroom consumption are accurate for the quantity of wild mushrooms consumed in Fukushima or not, and we added this as one of the limitations. (Page 13, Line 235-242).

119) "Fig 1. Distribution of concentrations of radiocesium(134Cs+137Cs) in wild mushroom samples collected in Kawauchi village (2016–2019)."

Comment: The activities are separated into only four bins: n.d; n.d.-100Bq; 100-1000Bq; >1000Bq. This appears consistent with the earlier papers, although Nakashima et al, 2015, uses five bins. It would be helpful to include more granular histogram for each year, with at least ten bins each, so the actual activity distributions could be better grasped.

According to your suggestion, we added more bins and replaced Fig 1.

141) "Table 2. Internal committed effective dose calculated with the assumption that residents consume wild mushrooms collected in Kawauchi village for 1 year."

Comment: Related to the above comment. It would be helpful to add the estimated quantity consumed for each age group to the table.

According to your suggestion, we added the mushroom intake information for each age group in the Table 2.

152) "The median concentration of radiocesium in all mushroom species was 526 Bq/kg in 2016, 88 Bq/kg in 2017, 1,106 Bq/kg in 2018 and 362 Bq/kg in 2019."

Comment: These are very wide fluctuations. You touch on some of the reasons for this later. Perhaps one finding is that the sampling methodology reported in this study is unable to provide representative measurements of the overall change in radiocesium levels in wild mushrooms in general in Fukushima. However, it can provide a good indication of typical intake of radiocesium from consumption of gathered wild mushrooms.

As you describe, we also think that this is a key point.

154) "These results suggest that the proportion of mushroom samples with radiocesium concentrations above 100 Bq/kg did not dramatically change in Fukushima over 8 years."

Comment: This is a key finding, and consistent with the literature regarding Chernobyl.

Thank you for your comments.

75) "We showed that the estimated committed effective dose from the ingestion of wild mushrooms collected in 2016 was 0.017 – 0.054 mSv, 0.0028 – 0.0087 mSv in 2017, 0.034 – 0.110 mSv in 2018 and 0.011 – 0.036 mSv in 2019. Previously, we reported that the calculated effective dose ranged from 0.1 – 1.6 mSv in 2013 [14] and from <0.001 to 0.6 mSv in 2015 [16]. These findings suggest that internal radiation exposure due to the consumption of wild mushrooms remained relatively limited in Kawauchi village."

Comment: These points as stated appear to be valid. However, as noted later, there is great variation in individual consumption of wild mushrooms in Fukushima, and the population has been repeatedly advised not to eat them. It is likely that these are large overestimates for the majority of the population, and underestimates for a small number of individuals who consume large quantities of the most contaminated varieties.

According to your comment, we added the sentence you mentioned above as a limitation of this study. (Page 13, Line 235-242.)

190) "Uncertainties arise because the committed effective dose from dietary intake of mushrooms does not capture day-to-day variation in individuals."

Comment: As noted above.

According to your comment, we added the sentence you propose as a limitation of this study. Thank you for your suggestions. (Page 11, Line 235-242.)

---

## [Editor Report · Decision Letter 1]

3 Sep 2020

Radiocesium concentrations in mushrooms collected in Kawauchi Village five to eight years after the Fukushima Daiichi nuclear power plant accident

PONE-D-20-19962R1

Dear Dr. Orita,

We’re pleased to inform you that your manuscript has been judged scientifically suitable for publication and will be formally accepted for publication once it meets all outstanding technical requirements. Congratulations!

Kind regards,

Tim A. Mousseau

Academic Editor

PLOS ONE
---

## [Editor Report · Acceptance letter]

7 Sep 2020

PONE-D-20-19962R1 

Radiocesium concentrations in mushrooms collected in Kawauchi Village five to eight years after the Fukushima Daiichi nuclear power plant accident 

Dear Dr. Orita:

I'm pleased to inform you that your manuscript has been deemed suitable for publication in PLOS ONE. Congratulations! Your manuscript is now with our production department. 

Kind regards, 

on behalf of

Dr. Tim A. Mousseau 

Academic Editor

PLOS ONE